# Microbiome Influence in the Pathogenesis of Prion and Alzheimer’s Diseases

**DOI:** 10.3390/ijms20194704

**Published:** 2019-09-23

**Authors:** Valeria D’Argenio, Daniela Sarnataro

**Affiliations:** 1Department of Molecular Medicine and Medical Biotechnologies, University of Naples Federico II, via Pansini 5, 80131 Naples, Italy; dargenio@ceinge.unina.it; 2CEINGE-Biotecnologie Avanzate, via G. Salvatore 486, 80145 Naples, Italy; 3Task Force on Microbiome Studies, University of Naples Federico II, 80131 Naples, Italy

**Keywords:** prion disease, Alzheimer’s disease, misfolded proteins, mutations, gut microbiota

## Abstract

Misfolded and abnormal β-sheets forms of wild-type proteins, such as cellular prion protein (PrP^C^) and amyloid beta (Aβ), are believed to be the vectors of neurodegenerative diseases, prion and Alzheimer’s disease (AD), respectively. Increasing evidence highlights the “prion-like” seeding of protein aggregates as a mechanism for pathological spread in AD, tauopathy, as well as in other neurodegenerative diseases, such as Parkinson’s. Mutations in both PrP^C^ and Aβ precursor protein (APP), have been associated with the pathogenesis of these fatal disorders with clear evidence for their pathogenic significance. In addition, a critical role for the gut microbiota is emerging; indeed, as a consequence of gut–brain axis alterations, the gut microbiota has been involved in the regulation of Aβ production in AD and, through the microglial inflammation, in the amyloid fibril formation, in prion diseases. Here, we aim to review the role of microbiome (“the other human genome”) alterations in AD and prion disease pathogenesis.

## 1. Introduction

Prion diseases, similar to Alzheimer’s and Parkinson’s diseases, are based on the aggregation of abnormal protein assemblies which could result in a cascade of neurodegenerative pathways. Prion diseases, whose origin can be sporadic, genetic, or infectious, are characterized by misfolding of a normal prion protein (PrP^C^) into a pathological form, called PrP^Sc^. Several germline, predisposing mutations have been associated with prion diseases pathogenesis; these mutations can affect the entire *PRNP* sequence length, including the signal sequence of the glycosylphosphatidylinositol (GPI)-anchor attachment [1]. It has been hypothesized that *PRNP* mutations are able to lower the energy barrier for the conversion of the normal cellular PrP into the scrapie, pathological PrP^Sc^ isoform [2]. The latter, according to the recently proposed concept of “propagating misfolding”, can modulate the misfolding of the normal protein into the misfolded amyloidogenic isoform [3].

Similar to the misfolded and aggregated PrP, other protein particles, deriving from the misfolding and aggregation of the Aβ or tau proteins, are able to propagate from affected cells to neighboring cells by a well-documented intercellular transfer of protein inclusions [4]. As well as PrP, mutations in the APP (amyloid precursor protein) cleavage sites and in the Aβ sequence can affect the processing of the protein with consequences in the trafficking, folding, and aggregation/amyloidogenic abilities [5,6].

The infective prion disease pathogenesis is based on protein misfolding triggered by prions entering the gut [7] where microbiomal factors (e.g., Curli proteins) can act as a template for amyloid fibril formation though a cross-seeding event [8]. This suggests that the gut microbiota may play a role in the pathogenesis of prion disease; similarly, it has been hypothesized that gut microbiota may influence both Alzheimer’s disease (AD) pathogenesis and the risk for AD [9].

Indeed, the human microbiota, i.e., the communities of microorganisms that live inside and on the surface of our body, has emerged as an important player for human’s health acquisition and maintenance, even before birth [10,11,12]. Consequently, a microbial dysbiosis has been associated with several diseases, suggesting new clues to clarify their pathogenesis [13,14,15,16,17,18,19]. Since microbiota composition is an actionable target, there is an increasing interest in understanding its functions and its possible use as a disease biomarker, or as a target for the development of novel therapeutic approaches [20,21,22,23]. Since it has been reported that there is a 1:1 ratio between human host and microbial cells, and that the microbial genes outnumber the host genes by about 150:1 [24,25], it is feasible to suppose that microbial genes features may contribute to AD and prion disease development, as well as human genome alterations.

Here, we analyze the latest findings on the role of the gut microbiota in the regulation of protein folding related to both prion and AD pathogenesis.

## 2. The Role of Microbiome in Prion and Alzheimer’s Disease

The gut microbiota has been proven to be crucial for several metabolic and immunological functions; in particular, influencing the gut–brain axis, the gut microbiota has been involved in brain development [26], neurogenesis [27], and specific interactions with the enteric and central nervous systems [28] (Figure 1A).

Despite a great interest, it is important to underline that most of the studies on human microbiota are observational and report just an association between a specific disease of interest and a microbial alteration. Currently, the challenge is to move to a more mechanistic view. In this context, different mechanisms are being investigated to clarify how the gut microbiota may influence the nervous system, including short-chain fatty acid secretion, blood–brain barrier permeability modifications, vagus nerve stimulation, or neurotransmitter modulation [29,30,31].

To date, gut microbiome alterations have been reported in different neurological disorders, including prion disease and AD [32,33,34,35,36,37,38,39,40,41,42,43].

Here, we will review the current knowledge regarding the role of the gut microbiome in these two neurodegenerative disorders.

### 2.1. Human Microbiota and Prion Disease Development

The pathological PrP^Sc^, a glycolipid-anchored sialoglycoprotein rich in β-sheets, is the causative agent of prion diseases, which can be of sporadic, genetic, or acquired origin [44]. Prion disorders are also referred to as TSEs (transmissible spongiform encephalopathies) and include the bovine spongiform encephalopathy (BSE) in cattle, scrapie in sheep, Creutzfeldt–Jacob diseases (CJD), Gerstman–Sträussler–Scheinker (GSS), fatal familial insomnia (FFI), and kuru in humans [45], which are characterized by a mechanism of protein misfolding adopted also by other neurodegenerative-related proteins, such as Aβ or tau [46].

Although 90% of prion diseases are sporadic, ~10% of all cases are due to dominantly inherited mutations in the prion protein gene *PRNP* [46]. In particular, mutations in *PRNP*, have been described to be present in the N-terminal region, in the central hydrophobic domain (HD), and in the C-terminal region including the GPI-attachment signal sequence (Figure 2).

These pathological mutations have been reported to have a key role in both the intracellular transport of secretory proteins and in the tendency to form β-sheet-enriched structures [47,48].

PrP intracellular localization and transport are of fundamental importance for the generation of the pathological PrP^Sc^ [49,50,51,52].

For some PrP mutants, it has been hypothesized that the mutations are able to lower the energy barrier for the conversion of the normal cellular PrP into the scrapie, pathological PrP^Sc^ isoform [2], and/or to induce an aberrant trafficking and accumulation inside the cells, triggering abnormal interaction with other unknown cofactors [52].

The gathering of the PrP^Sc^ in the brain of the affected subjects causes a reactive microglial and astroglial inflammation, and a consequent neurodegeneration. As for other diseases, it has been suggested that quantitative and/or qualitative alterations of the gut microbiota may play a role in prion-disease pathogenesis and susceptibility. In particular, it has been hypothesized that the prion agents ingested with food may cause a microbial dysbiosis at gut level. This dysbiosis in turn, through the production of a microbial form of amyloid, is able to activate the immune system, thus enhancing microglia and astrocyte activation and improving the production and accumulation of the neuronal amyloid at the brain level (Figure 1B) [43]. Items of evidence supporting this hypothesis are summarized below.

The interplay between the microbiota and the host immune system is crucial for immune system development. It has been reported that the gut microbiota is able to exert effects on the development of isolated lymphoid follicles [53,54]. Donaldson et al. showed that isolated lymphoid follicles were reduced in the small intestine of germ-free mice, and that their abundance increases after microbial colonization [55]. Since it has been reported that isolated lymphoid follicles are important sites of prion accumulation in the small intestine [55,56], this suggests that, acting on the microbiota, it is possible to: (1) reduce isolated lymphoid follicles density and, as a consequence, reduce the sites of prion uptake, replication, and neuroinvasion; and (2) reduce the development of prion disease pathogenesis [43]. The current opinion on BSE and kuru pathogenesis is based on the protein misfolding which is triggered by prion agents’ entry via the gut [7]. As the gut hosts the majority of microbiota, the major question is whether there are specific factors triggering the disease. Interestingly, it has been proposed that neuronal amyloid formation may be enhanced by exposure to microbiomal amyloids through cross-seeding (e.g., Curli proteins from different bacterial species that may serve as a template for fibril formation), which leads to misfolding of neuronal proteins in the brain in an analogous manner to kuru and BSE [8].

At the brain level, PrP^Sc^ accumulation causes an inflammatory reaction featured by microglia and astrocyte activation. It is well known that microglia are involved in neuronal homeostasis, synaptic remodeling, and defense against pathogens [57,58,59,60]. Microglia activation is one of the first sign of prion disease and is characterized by increased expression of the anti-inflammatory cytokines, such as transforming growth factor-beta (TGF-β) and prostaglandin E2 (PGE2), and signaling receptors, such as triggering receptor myeloid 2 cells (TREM2), sialic acid binding Ig-like lectin F (SiglecF), cell surface transmembrane glycoprotein CD200 receptor 1(CD200R), and fragment crystallizable gamma receptor (Fcg) [61,62]. This suggests that microglia activation should have a protective role with respect to the development of prion-related damage [57]. Accordingly, studies on MFGE-8 (milk fat globule—epidermal growth factor—factor VIII)- or cluster of differentiation 14 (CD14)-deficient mice showed an accelerated progression of prion disease with respect to the controls [58,63].

As a consequence, the sensing of bacterial lipopolysaccharide (LPS) may change this microglial protective status to a pro-inflammatory one. Indeed, Erny et al. have assessed that microglial development depends on gut microbiota [29]. They were able to demonstrate that in the brain of germ-free mice, there was an impaired microglial maturation due to reduced early exposure to LPS of the commensal microbiota, and a similar behavior was observed also in SPF (specific pathogen free) mice treated with broad-range antibiotics [29]. Thus, it is conceivable that prolonged use of antibiotics during prion diseases, by impairing the gut microbiota, may hamper the anti-inflammatory activity of the microglia and negatively impact on disease development.

Moreover, diet has been reported as a factor able to affect microglial development and functions through gut microbiota alterations. Indeed, it has been established that Bacteroidetes and Clostridia are able to metabolize the polysaccharides in dietary fibers into short-chain fatty acids (SCFAs). Among the SCFAs, butyrate not only regulates gut mucosal permeability and thus is involved in inflammatory response, but also seems to be related to microglial homeostasis [29,64]. Consequently, a high-fat diet (including the Western diet), being able to increase the Firmicutes level and reduce the quantity of Bacteroidetes, could reduce the availability of SCFAs important for microglial activity and promote prion diseases. However, this association has not been confirmed to date.

### 2.2. Human Microbiota and Alzheimer’s Disease Development

Similarly to prion diseases, a minority of Alzheimer’s disease cases is due to dominantly inherited mutations affecting mainly the *APP* gene. For *APP*, the critical sequences involved in aminoacidic mutations, affecting folding, dynamic protein stability, and proteins aggregation, are in the Aβ and transmembrane regions (Figure 2).

It has been estimated that about 40 million people worldwide are affected by AD (WHO, World Health Organization), being the most common kind of dementia. AD is a neurodegenerative disorder featured by a chronic neuroinflammation (increased levels of IL-1β, TNF-α, COX-1, COX-2, 12-LOX, and 15-LOX) due to an anomalous activation of the microglia, especially in the area of the brain (usually, the frontal cortex and hippocampus) that shows a higher concentration of extracellular Aβ plaques and intracellular neurofibrillary tangles (NFTs) [65].

Aging is associated with an increasing risk for developing inflammatory-based diseases, such as generalized cognitive impairments and neurological conditions, including AD. Indeed, aging not only is characterized by molecular and cellular modification of the human host, but also by modifications of the gut microbiome composition [66]. The role of bacteria in lifespan development, aging, and dementia is still largely unknown. However, it has been assessed that elderly individuals with low bacterial diversity had lower cognitive functions [67]. Even if cause–effect relationships are yet to be proven, this observation opens the way for future investigations [66].

An emerging hypothesis is that the gut microbiota may influence AD neuroinflammation. In particular, it has been proposed that in AD, similarly to prion diseases, a microbial dysbiosis at the gut level may impair gut permeability, thus inducing a systemic activation of the immune system. The latter, together with the production of bacterial Aβ fibrils, is able to enhance the neuroinflammatory status and the deposition of the Aβ fibrils at the brain level (Figure 1C).

Accordingly, Harach et al. highlighted the presence of a microbial dysbiosis at gut level in *APP* transgenic mice with respect to the wild type [68]. Next, they assessed that the colonization of germ-free *APP* transgenic mice with the gut microbiota of *APP* transgenic mice was able to enhance Aβ-related cerebral damage, while the colonization with the gut microbiota from wild-type mice had a poor effect on Aβ levels in brain [68]. Thus, these data strongly support the idea that the gut microbiota may contribute to AD development.

Furthermore, it has been assessed that some high-fat diets increase the risk for AD, and this seems to be related to alterations of gut permeability, at least in mice [9,69]. Thus, it is possible to hypothesize that such diets increase the risk for AD by inducing specific modifications of the gut microbiota composition [65].

According to this hypothesis, Cani et al. found that the gut microbiome of mice fed with a diet enriched in unsaturated fatty acids, had a reduction of *Lactobacillus*, *Bacteroides*, and *Prevotella* species and an increase of *Bifidobacterium* species with respect to that of the control group [70]. They also found that the treated group had an increase of gut permeability, probably due to the reduced expression of both zonulin and occludin proteins, corresponding to higher circulating level of the bacterial LPS and also of inflammatory markers, such as IL-1 and TNF-α. Interestingly, oral antibiotic administration was able to restore a gut microbiota similar to that of the control group and, by increasing the expression of both zonulin and occludin proteins, to restore also the gut permeability [70]. Subsequently, Zhang et al. showed that AD patients had increased levels of serum LPS and monocyte activation with respect to the controls [71]. Since it is known that in AD the blood–brain barrier is damaged and this can lead to the anomalous microglia activation and to the typical AD neuroinflammation, the combination of an altered permeability also at intestinal level may allow a pathogenetic signaling between the gut microbiota and the central nervous system. In particular, the microbial dysbiosis altering gut permeability is able to trigger a systemic inflammatory status that may potentiate the neuroinflammatory reactions typical of AD development [65,72]. In this context, Zhao et al. reported an increased concentration of the bacterial LPS in the hippocampus and neocortex of AD patients’ postmortem brains [73]. It is noteworthy that patients affected by inflammatory bowel diseases, showing an altered gut permeability due to gut microbiota alterations, are at high risk for AD [74,75].

Furthermore, Cattaneo et al. highlighted a relationship between cognitive impairment, brain amyloidosis, and the presence of circulating inflammatory markers [76]. They found that cognitively impaired AD patients had higher circulating levels of pro-inflammatory cytokines (IL-6, CXCL2, NLRP3 and IL-1β) and lower levels of anti-inflammatory cytokine (IL-10), and that the *Escherichia/Shigella* amount correlated positively with this behavior; consequently, they hypothesized that alterations of the gut microbiota, i.e., increased levels of pro-inflammatory (*Escherichia*/*Shigella*) and reduced levels of anti-inflammatory (*Eubacterium rectale*) species, may have a role in AD cognitive impairment [76].

A recent study by Haran et al. carried out a five-month prospective study collecting longitudinal stool samples for metagenomic analysis and intestinal cell functional evaluations in AD patients compared to elderly without dementia or with other kinds of dementia [77]. This study showed that the microbiota of the AD patients had a higher abundance of pro-inflammatory taxa and a lower abundance of butyrate-producer taxa compared with the other two study groups. In addition, they were able to verify that the AD-related microbiota alters the P-glycoprotein (a mediator of gut homeostasis) pathway, supporting the hypothesis that an altered epithelial homeostasis is a mechanism by which the microbiome impacts AD pathogenesis [77].

An intriguing pathogenetic hypothesis, recently published by Block, suggests a cooperative model in which some pathogenic microbes may enhance the virulence of others, usually less invasive, enabling them to cross the blood–brain barrier and exert pathogenic affects at the brain level [78]. The authors, in particular, hypothesize that a microbial dysbiosis at the gut level together with involving *Chlamydia* species and Human papillomavirus infections, altering the mucosal barrier integrity, allows colonization from yeast. In particular, *Chlamydia* and other gut microbiota commensals induce insulin resistance, thus providing the glucose required for yeast growth, and also predispose to amyloid beta protein accumulation. In addition, immunological alterations due to both the papilloma virus infection and the *Chlamydia*-induced hyperglycemia enable the invasion of the brain by yeast. At the brain level, inflammatory cytokines produced in response to yeast increase amyloid beta protein accumulation [78].

In addition to the effect on gut permeability, it has been proposed that the gut microbiota may also have a direct effect on some specific AD biomarkers [65]. Indeed, through a metabolomics screening, Xu and Wang (2016) highlighted a positive correlation between a pattern of about 50 microbial metabolites and AD early age of onset and cognitive decline [79]. Interestingly, among the increased metabolites, there was also TMAO (trimethylamine N-oxide), a microbial metabolite associated with the assumption of animal fats, thus supporting the hypothesis that high-fat diets may be a risk factor for AD development also by inducing a gut microbial dysbiosis, which in turn exerts a series of pro-inflammatory, pathogenetic functions.

It is also noticeable that different studies have reported the ability of different bacteria, such as *Salmonella enterica* and *typhimurium*, *Mycobacterium tuberculosis*, *Staphylococcus aureus*, *Bacillus subtilis*, and *E. coli*, to produce the amyloid protein [80,81,82]. In this context, Asti and Gioglio (2014) assessed that *E. coli* LPS increases Aβ fibril production [83]. Since it is known that Aβ fibrils pass through both gut and brain barriers [84,85,86], an intriguing hypothesis is that different bacteria in the gut may produce amyloid fibers that are able to activate the immune system and, thus, trigger the typical AD brain neuroinflammation [87]. Holmqvist et al. also hypothesized that the Aβ fibrils produced by gut bacteria may be directly deposited into the brain promoting AD [88]. In particular, *Porphyromonas gingivalis*, the etiologic factor of chronic periodontitis, has caught the attention since not only was it identified in the brain of AD patients, but its toxic proteases, namely gingipains, were also found in AD patients’ brain [89]. Interestingly, *P. gingivalis* infection in AD mice models has been proven to cause brain colonization and an increased production of the Aβ1–42 [90]. In addition, gingipains have showed neurotoxic effects both in vivo and in vitro, impairing tau and consequently the normal neuronal function [91]. Based on the hypothesis that *P. gingivalis* infection, through gingipains production, is able to promote AD-typical neuronal damage, a recent work by Dominy et al. tested the use of a gingipains inhibitor in positively modulating AD progression [92]. Interestingly, they found that gingipain inhibition was able to reduce *P. gingivalis* brain infection, inhibit Aβ1–42 production, ameliorate the neuroinflammatory status, and protect neurons in the hippocampus. These results suggest that the use of gingipain inhibitors could be useful to treat *P. gingivalis* brain colonization and ameliorate neurodegeneration in AD patients [92].

Bacteria seem to have a role also in the expression of the triggering receptor expressed on myeloid cells-2 (TREM2). This receptor is expressed on the surface of microglial cells and has been involved in several functions, including cytokines and ROS production, phagocytosis, and Aβ clearance [93,94]. It has been reported that TREM2 is down-expressed in AD and that the bacterial LPS can hamper the expression of this receptor reducing, in turn, the phagocytosis of the Aβ fibrils [95,96,97].

The latter data support the idea that gut microbiota alterations may influence AD development, and that a combination of different mechanisms is involved in this process. Bacteria not only seem to be able to produce and/or induce the production of Aβ fibrils, but also to contribute to their accumulation at brain level.

A recent study reported for the first time an association between the gut microbiota, bile acid profile, and genetic variants in AD pathogenesis [98]. They found that the serum bile acid profile is altered in AD patients with a significant decrease of liver-derived primary bile acids and an increase of bacterially produced secondary bile acids and their conjugated forms compared to control subjects. The higher levels of secondary conjugated bile acids were significantly associated with worse cognitive function. In addition, several genetic variants in immune-response-related genes, previously associated with AD, seem to be associated with bile acid profiles [98].

Interestingly, it has been highlighted that some bacteria may exert a positive effect reducing AD-related signs. Wang et al. showed that in rats fed with grape seed polyphenol extracts there is an accumulation in the brain of polyphenol metabolites that are able to interfere with Aβ oligomerization [99]. Other studies have assessed that *Lactobacillus* and *Bacillus* species produce acetylcholine, a neurotransmitter, usually reduced in the brain of AD patients and, thus, may have a beneficial effect [100,101].

In addition to the effects on Aβ fibrils, it is important to underline that gut microbiota has also been implicated in some cognitive functions and in memory [102]. Pyndt Jorgensen et al. showed that mice fed with a high-fat (especially saturated) diet had specific microbial modifications (increased levels of the Firmicutes phylum and *Rumunococcus* genus, and reduced level of the Bacteroidetes phylum) that correlated with a reduction of memory [103].

An intriguing study showed in a mouse model of AD (APPSWE/PS1ΔE9) that alterations of the gut microbiota induced by antibiotics administration affect both inflammation and amyloid production [104]. In particular, the mice treated with antibiotics had increased levels of Lachnospiraceae and decreased levels of *Bacteroides* with respect to the untreated, and these microbial alterations correspond to a lower plaque burden [104].

Recently, Abraham et al. studied the effects of regular physical exercise and nutritional intervention on the development of AD in APP/PS1 transgenic mice [105]. Interestingly, they showed that exercise, probiotics, and their combined effects reduced the number and area of amyloid plaques in the hippocampus of mice, and that these beneficial effects were partly due to alterations of the gut microbiome [105]. In this context, Akbari et al., conducting a randomized, double-blind, controlled trial, showed that AD patients treated with 200 mL/day of probiotic (three strains of lactobacilli and *Bifidobacterium bifidum*) for twelve weeks experienced benefits in cognitive function and metabolic state [106]. Indeed, several recent studies have been carried out to prove the hypothesis that pro- and prebiotics may have beneficial effects in ameliorating AD-typical alterations [107,108,109,110,111]. In this context, Leblhuber et al. showed that the supplementation of AD patients with a multispecies probiotic modifies the gut bacteria composition and also tryptophan metabolism in serum, suggesting a role in immune system activation [107]. Similarly, Athari Nik Azm et al. found that multispecies probiotic administration is able to improve insulin resistance in an experimental model of AD [108].

On the contrary, Agahi et al. conduced a double-blind clinical trial to evaluate the effects of 12 weeks of probiotic treatment on inflammatory and oxidative biomarkers, and they did not find any significant difference [109]. This result suggests that, in addition to the kind and dosage of probiotic bacteria, AD severity and time of administration affect treatment outcomes.

In an attempt also to standardize the methodologies for probiotic administration, Sanborn et al. are carrying out a randomized clinical trial to evaluate the potential effects of three months of LGG (Lactobacillus rhamnosus GG) probiotic supplementation on psychological status and cognitive performance in middle-aged and older adults; their results will provide further insights on these aspects [110].

Finally, Asl et al. have recently assessed the effects of probiotic administration in an animal model of AD on behavioral and electrophysiological aspects [111]. They were able to demonstrate that, while the control group experienced a weak spatial performance, the probiotic-treated group improved the maze navigation. In addition, probiotic administration significantly restored the long-term potentiation and exerted a positive effect on the antioxidant/oxidant biomarker balance, thus suggesting that probiotics have a positive effect on synaptic plasticity [111].

In addition to probiotic administration, also other nutritional interventions and/or dietary supplementations are being investigated to assess their role in AD pathogenesis and, consequently, their possible use for therapeutic purposes [112]. In this context, the role of tryptophan has been recently reviewed, since it has been suggested that the gut microbiota may play an important role in tryptophan availability regulation [113]. Indeed, tryptophan is an essential amino acid involved in emotional states, cellular aging, and serotonin pathway, and it has been shown that some bacteria in the gut may consume the food-derived tryptophan, therefore, gut microbiota manipulation may be used as a strategy for regulating the tryptophan availability and/or to reduce the concentration of some neurotoxic derivates, lowering the severity of neurodegenerative disorders, such as AD [113].

Nagpal et al. have evaluated, in subjects with mild cognitive impairment with respect to controls, the effects of a modified Mediterranean ketogenic diet on the gut microbiota composition in association with other AD biomarkers [114]. They not only found specific microbial alterations positively correlated with mild cognitive impairment, but also highlighted that the proposed ketogenic diet is able to modulate the gut microbiota and induce improved AD-related biomarkers in the cerebrospinal fluid [114].

In a recently published study, McCann et al. explored the association between gut microbiota-derived vitamin K and cognitive function [115]. Indeed, vitamin K has protective effects on myelin integrity and has been found to be reduced in AD patients. Since some commensal bacteria are able to produce vitamin K, this may be a mechanism though which microbial dysbiosis may exert pathogenic effects. Interestingly, by correlating the gut metagenomes of the elderly with their different cognitive levels, some vitamin K isoforms were identified as positively correlated with cognition [115].

Taken together, all these data highlight the role of the gut microbiota in AD pathogenesis through different mechanisms, synergistically contributing to the development of the disease-specific signature.

## 3. Conclusions

The vicious circle of spread, seed, and accumulation of misfolded protein aggregates within the central nervous system is not only restricted to prion protein PrP, but has also been reported for other proteins, such as Aβ, α-synuclein, and tau, that are able to form amyloidogenic aggregates which spread through the brain and cause distinct neurodegenerative diseases [3,45]. Mutations in the genes codifying for these proteins have been associated with the pathogenesis of prion diseases and AD, and have been described to alter protein trafficking, processing, dynamic stability, folding, and ability to aggregate inside the cells [5,6].

Indeed, defects in APP intracellular trafficking and processing leading to Aβ production in neurons, as well as defects in PrP metabolism, are likely the most common cause of AD and prion diseases, respectively; thus, understanding how APP and PrP are targeted to a selected destination inside the cells and identifying the molecular mechanisms controlling the Aβ /prions generation is a key for new therapies.

Interestingly, regulation of Aβ production and neuroinflammation has been described to be enhanced by gut microbiota in AD. Increasing evidences are also suggesting a role of the microbiome in microglia activation typical of prion diseases and beside the critical role of the interplays between the host and its gut microbiota in the availability of micronutrients, trophic factors, and neurotransmitters, it has been reported to also have a key role in the development of cognitive and behavioral functions (known to be compromised in disease conditions). Accordingly, alterations of the gut microbiota have been associated with AD and prion diseases [9] and seem to be able to contribute to functional impairment, neuroinflammation, and exacerbation of disease-specific pathogenic reactions.

Although the data obtained up to date are still preliminary and need to be further confirmed in controlled clinical trials, they draw attention to an emerging field of neuroscience that may acquire an increasing relevance in a near future.

## Figures and Tables

**Figure 1 ijms-20-04704-f001:**
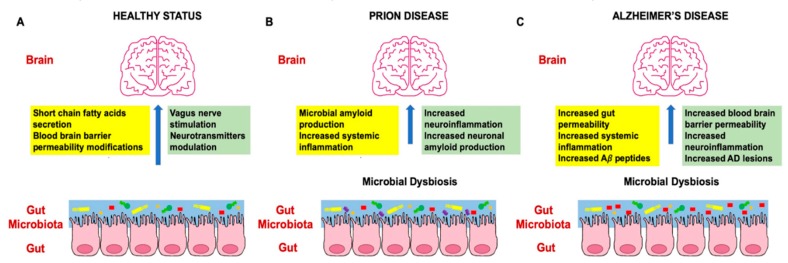
Relationship between the gut microbiota and the brain. (**A**) In physiological conditions, gut microbiota exerts its effects on the central nervous system using several mechanisms, including the production of short-chain fatty acids, modification of blood–brain barrier permeability, modulation of specific neurotransmitters, and vagus nerve stimulation. (**B**) In prion disease, prion agents introduced with diet give a microbial dysbiosis that, through microbial amyloid production, is able to activate the immune system, enhance microglia and astrocyte activation, and improve neuronal amyloid production and accumulation in the brain. (**C**) In Alzheimer’s disease, the microbial dysbiosis impairs gut permeability, inducing a systemic activation of the immune system; this, together with bacterial amyloid beta (Aβ) fibrils, increases the neuroinflammatory status and the deposition of Aβ fibrils at the brain level.

**Figure 2 ijms-20-04704-f002:**
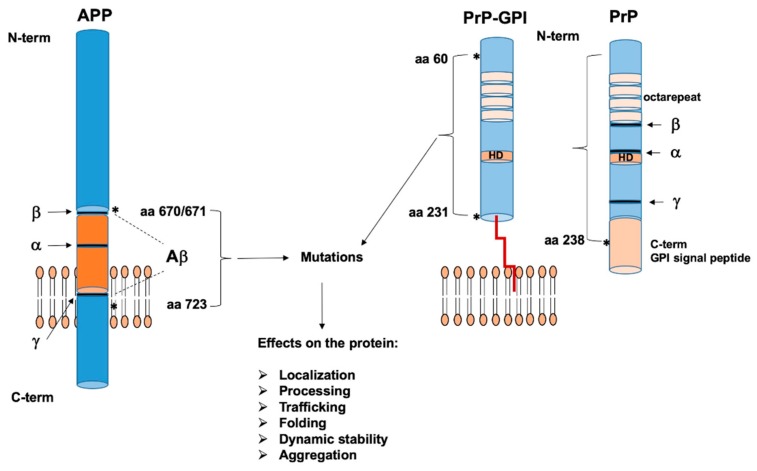
Schematic illustration of Aβ precursor protein (APP) and prion protein (PrP) pathogenic mutations. Sites of α -, β-, and γ-secretase cleavage are indicated by arrows on both proteins. Asterisks near braces point to currently known mutations in APP and PrP: from aa670/671 to aa723 in APP; from aa60 to aa231 in glycosylphosphatidylinositol (GPI)-anchored PrP and to aa238 (present in the GPI-attachment signal) in unprocessed PrP (carrying the GPI-signal peptide). To note: APP can be principally mutated into or nearby the Aβ sequence, while PrP can be mutated all along the sequence length. Effects of protein mutations are listed in the column. HD: hydrophobic domain.

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
