# Peer review of "Microbiome Influence in the Pathogenesis of Prion and Alzheimer’s Diseases"

_ijms, 2019, doi:10.3390/ijms20194704_

Round 1

Reviewer 1 Report

This review has been revised and the authors have taken into account the minor comments made. However, as per my original review of this manuscript, I still feel that the authors have tried to cover too large an area in one review article and there is no clear link. At present they do not give justice to the section on gene/mutations in prion disease and AD. This area is frequently covered by reviews and sadly this review is not adding anything to the literature either in terms of data or perspective.

The interesting aspect of this manuscript is the second section on the microbiome and prion disease/AD of which there are few current reviews.  This section should be the focus of the review.

I recommend that review is revised to remove the gene/mutations sections and concentrate only on the microbiome areas. This could include a brief summary of the uptake of PrPSc through the gut and the microbiome areas could be expanded even more.

Author Response

According to the Reviewer’s suggestion, we modified the manuscript removing the section on the gene/mutations and focusing on the microbiome areas. We have also added some recent papers to the discussion regarding AD pathogenesis and gut microbiota (see the revised manuscript pag. 6, lines 1-19; and the novel references #78-79). Finally, we provide now some clues related to gut microbiome and diet in Alzheimer disease (see the revised manuscript pag. 8, lines 5-26; and the novel references #113-116).

Reviewer 2 Report

The review has been extensively revised and is now much improved.  It now much clearer and provides a more in depth critical analysis.  

It does require some changes in the English.  

Author Response

We thank the Reviewer for the positive comments. The manuscript has been revised for English style revisions.

Round 2

Reviewer 1 Report

This manuscript is now much stronger than the previous versions and is a nice review of the microbiome influence in prion disease and AD.

I have a few minor comments:

Change title to remove 'genetic determinants'  p2, line 37: 'state of art', change to 'current knowledge' pg 3, line 4; remove 'transmissible' before 'bovine' pg 3, line 9; change 'even if' to 'although' and remove the specific examples as some of those diseases have both sporadic and genetic forms' pg 4, line 13: add 'entry via the gut' pg 6, line 9; remove 'Janice' REFERENCES: the references are out of sync between text and ref list and need to be double checked 

Author Response

This manuscript is now much stronger than the previous versions and is a nice review of the microbiome influence in prion disease and AD.

We thank the Reviewer for this positive comment.

I have a few minor comments:

Change title to remove 'genetic determinants'

Done.

p2, line 37: 'state of art', change to 'current knowledge'

Done.

pg 3, line 4; remove 'transmissible' before 'bovine'

Done.

pg 3, line 9; change 'even if' to 'although' and remove the specific examples as some of those diseases have both sporadic and genetic forms'

Done.

pg 4, line 13: add 'entry viathe gut'

Done.

pg 6, line 9; remove 'Janice' 

Done.

REFERENCES: the references are out of sync between text and ref list and need to be double checked

Done.